# Estimating and Penalizing Induced Preference Shifts in Recommender Systems

## Abstract

The actions that a recommender system (RS) takes – the content it exposes users to – influence the preferences users have over what content they want. Therefore, when an RS designer chooses which system to deploy, they are implicitly *choosing how to shift* or influence user preferences. Even more, if the RS is trained via long-horizon optimization (e.g. reinforcement learning), it will have incentives to manipulate preferences, i.e to shift them so they are more easy to satisfy, and thus conducive to higher reward. While some work has argued for making systems myopic to avoid this issue, myopic systems can still influence preferences in undesirable ways. In this work, we argue that we need to enable system designers to *estimate* the shifts an RS *would* induce; *evaluate*, before deployment, whether the shifts are undesirable; and even *actively optimize* to avoid such shifts. These steps involve two challenging ingredients: *estimation* requires the ability to anticipate how hypothetical policies would influence user preferences if deployed – we do this by training a user predictive model that implicitly contains their preference dynamics from historical user interaction data; *evaluation* and *optimization* additionally require metrics to assess whether such influences are manipulative or otherwise unwanted – we introduce the notion of "safe shifts", that define a trust region within which behavior is believed to be safe. We show that recommender systems that optimize for staying in the trust region can avoid manipulative behaviors (e.g., changing preferences in ways that make users more predictable), while still generating engagement.

## 1 Introduction

A recommender system (RS) exposes users to feeds of items, and users choose what to click on based on their preferences. These preferences are non-stationary: they shift over time (Koren, 2009; Rafailidis & Nanopoulos, 2016; Li et al., 2014) . Importantly, these shifts are not independent from the RS's actions: what content the RS exposes users to influences their preferences.

Because of this influence, when a system designer chooses a specific recommender algorithm (policy), they are implicitly choosing how to shift user preferences. A growing trend is to choose policies that stem out of long-term optimization of user engagement (Afsar et al., 2021) – typically via reinforcement learning; we will refer to these as long-term value, or LTV, systems. However, these policies might lead to very undesirable shifts as a side-effect: certain preferences are easier to satisfy than others, leading to more potential for engagement – this could be because of availability of more content for some preferences compared to others, or because very strong preferences for a particular type of content lead to higher engagement than more neutral ones. The LTV system therefore has an incentive to drive user preferences over time into these areas of higher engagement potential. This can make such LTV systems a particularly poor choice for avoiding undesired shifts.

While it has been proposed to stifle RSs' capabilities by making them myopic – to avoid the incentives for preference manipulation above (Krueger et al., 2020) – even myopic systems can influence preferences in systematically undesirable ways (Jiang et al., 2019; Mansoury et al., 2020; Chaney et al., 2018). Fig. 1 shows a simple example, plotting user preferences in a 1D space over time (in a simulated environment). On the right, we see how interacting with the RL policy drives users strongly away from their initial preferences to a specific small region. The myopic policy (center), shows the same type of effect, but much less pronounced, where after 10 time steps some users do reach other preferences as well.

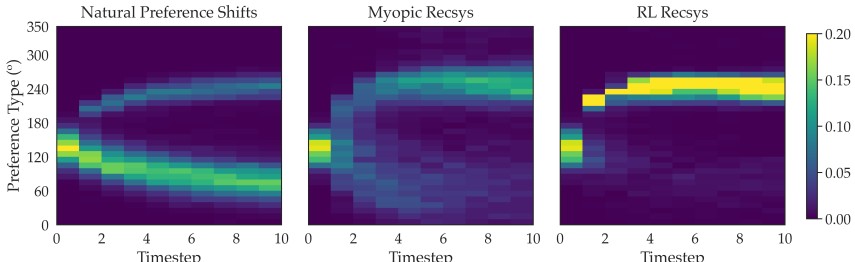

Figure 1: **Preferences induced by different RS policies across the same cohort of users.** In simulated environment, preferences (y axis) are in 1D, and change over time (x axis) as users interact with the policy. On the right, the RL system drives preferences to one spot. A myopic policy (center) has a similar effect but less concentrated. These shifts are different from what we introduce as the "natural" evolution of user preferences.

The reality is that any policy will have influence on user preferences. In this work, we argue that *we need to explicitly account for this influence when choosing which policy to deploy.* This requires tackling two critical problems. First, we need to anticipate what preference shifts an RS policy will cause *before it is deployed*. This alone would enable system designers to qualitatively assess whether a policy induces shifts that seem manipulative or otherwise unwanted. We study how to do this from historical user interaction data of previous RS policies, and show how to train a predictive model that implicitly captures preference dynamics. The second critical problem is having metrics for *quantitatively* evaluating whether an evolution of preferences is actually unwanted: a computable metric not only simplifies evaluation, but also enables RSs to *actively optimize against unwanted shifts*. Unfortunately, defining what "undesirable" shifts are is a complex philosophical and ethical challenge (noa, 2018; Paul, 2014). To side-step it, we propose a conservative approach based on the notion of "safe shifts": instead of defining what is undersirable, we non-exhaustively define certain shifts that we trust not to be problematic, and measure the extent to which a policy deviates from them. For instance, a candidate for safe shifts that we introduce is how user preferences would *naturally* shift if the user were "omniscient", i.e. if they could see and attend to the true distribution of available content without any bias from the RS selecting specific slates to show them. Fig. 1 (left) shows this natural preference evolution for our running example, and how user preferences stay somewhat diffuse but drift towards the opposite mode that the RL and myopic policies push them to. Note that while this metric can effectively penalize undersired shifts, it comes at a cost: natural shifts, and in fact any lists of safe shifts that we define, are unlikely to be exhaustive, which means the approach will conservatively penalize policies that might in reality be safe.

To demonstrate both the estimation of preference shifts and their evaluation, we set up a testing environment in which we emulate ground truth user behavior by drawing from a model of preference shift from prior work (Bernheim et al., 2021). We first show qualitatively that in this environment, RL and even myopic RS lead to undesired shifts. We then find that our preference shift estimation model, if trained based on historical user interaction with different policies, can anticipate these shifts (see Fig. 7). Further, we find that our evaluation metric correctly flags which policies will produce undesired shifts, and evaluates the RL policy from Fig. 1 as $\tilde{3}5\%$ worse than the myopic one, which is in turn $\tilde{4}0\%$ worse than a policy which avoids shifting preference to the "unnatural" mode from the figure. Our results also suggests that evaluating our metric using the trained estimation model correlates to using ground truth preference dynamics, and that optimizing for safe shifts does lead to higher scoring (more safe) policies.

Although this just scratches the surface of finding the right metrics for unwanted preference shifts and evaluating them, our results already have implications for the development of recommender systems: in order to ethically use systems at scale, we e active steps to *measure* and *penalize* how such systems shift users' internal states. In turn, we offer hope that this is possible – at least for preference shifts – by learning from user interaction data, and put forward a framework for specifying "safe shifts" for detecting and controlling such incentives.

## 2 RELATED WORK

**RS effects on users' internal states.** A variety of previous work considers how RSs can affect users: influencing user's preferences for e-commerce purposes (Häubl & Murray, 2003; Cosley et al., 2003; Gretzel & Fesenmaier, 2006), altering people's moods for psychology research (Kramer et al., 2014), changing populations' opinions or behaviors (Matz et al., 2017), exacerbate (Hasan et al., 2018) cases of addiction to social media (Andreassen, 2015), or increase polarization (Stray, 2021). There

have been three main types of approaches to quantitatively estimating *effects of RSs' policies* on users: 1) analyzing static datasets of interactions directly (Nguyen et al., 2014; Ribeiro et al., 2019; Juneja & Mitra, 2021; Li et al., 2014), 2) proposing hand-crafted models of user dynamics and using them to simulate interactions with RSs (Chaney et al., 2018; Bountouridis et al., 2019; Jiang et al., 2019; Mansoury et al., 2020; Yao et al., 2021), or 3) using access to real users and estimating effects through direct interventions (Holtz et al., 2020; Matz et al., 2017). Our approach is most similar to 2), but we propose implicitly *learning* user dynamics instead of handcrafting our own model.

**Neural networks for recommendation and human modeling.** While data-driven models of human behavior have been used widely in real-world RS as click predictors (Zhang et al., 2019; Covington et al., 2016; Cheng et al., 2016; Okura et al., 2017; Mudigere et al., 2021; Zhang et al., 2014; Wu et al., 2017) and for simulating human behavior in the context RL RSs' training (Chen et al., 2019; Zhao et al., 2019; Bai et al., 2020), to our knowledge they have not been previously used for simulating and quantifying the effect of hypothetical recommender systems on users. We emphasize how human models can also be used as auditing mechanisms by anticipating RSs' policies impact on users' *behaviors* and, as our method shows, even *preferences*.

**RL for RS.** Using RL to train LTV RSs has recently seen a dramatic increase of interest (Afsar et al., 2021), with some notable examples being proposed by YouTube and Facebook (Ie et al., 2019; Chen et al., 2020; Gauci et al., 2019) – which reported significant engagement increases with real users. Singh et al. in particular also takes a penalized RL approach.

**Side effects and safe shifts.** Our work starts from a similar question to that of the side effects literature (Krakovna et al.; Kumar et al., 2020), but applied to preference change: given that the reward function will be misspecified, how do we prevent undesired preference-shift side effects? The choice of safe shifts corresponds to the choice of baseline in this literature.

## 3 PRELIMINARIES

**Setup.** We model users as having time-indexed preferences $u_t \in \mathbb{R}^d$, which assign a scalar value to every possible item of content $x \in R^d$. and the user engagement derived from the consumption of item $x_t$ under $u_t$ at time $t$ is modeled as being given by $\hat{r}_t(u_t) = u_t^T x_t$. These preferences are part of the user's internal state $z_t$, along with other variables, which together with preferences comprise a sufficient statistic for their long-term behavior, but which our method will not explicitly model. At every time step, the user sees a slate $s_t$ produced by that RS policy $\pi$, and chooses an item $x_t$. The policy $\pi$ maps history of slates and choices so far, $s_{0:t}, x_{0:t}$, to the new slate $s_{t+1}$. Upon making a choice, the user's internal state updates to $z_{t+1}$.

**User choice model.** We are interested in inferring and predicting preferences, which we do not directly observe. As such, we make an assumption about how user behavior – their choice $x_t$ – relates to their current preferences $u_t$. Following prior work (Ie et al., 2019; Chen et al., 2019), we assume that users choose items in proportion to (the exponent of) the engagement under their current preferences, i.e. that $P(x_t = x | s_t, u_t)$ is given by the conditional logit model (detailed in Sec. 6).

**Casting this as an NHMM.** The dynamics of user internal state – which we don't assume to be known – depend on the history so far because the policy $\pi$ uses history. This makes our setup a non-homogeneous HMM (Hughes et al., 1999; Rabiner), with hidden state $z_t$ and dynamics that time-dependent: $P\big(z_{t+1}|z_t, \pi(s_{0:t}, x_{0:t})\big)$. *In our method* will not commit to an explicit representation of $z_t$, and will only implicitly learn these dynamics. *For our experiments* however, we will build a testing environment that does commit to a representation of $z_t$ via preferences and user beliefs, and adopts a "ground truth" dynamics of this internal state based on prior literature.

## 4 ESTIMATING USERS' PREFERENCES

Our proposal boils down to the following: before deploying a new recommender system policy $\pi'$, we first need to be able to understand how that policy would change preferences and behavior. In estimating how users' preferences would shift, one approach is to consider users which interacted with some policy $\pi$ (or a few policies), and try to estimate what preferences shifts $\pi'$ would induce in users if we were to deploy it *going forward*; this is useful, but suffers from a problem – users have already been biased by the previous policies, and their preferences might have already e.g. shifted to extremes; forward prediction will thus miss certain undesired effects. What we really want to do is turn back time – we want to ask for a new distribution of users identical to ours, how their

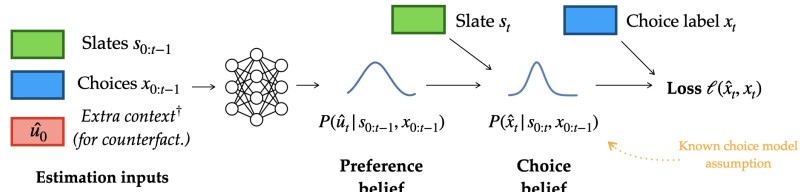

Figure 2: **Future preference estimation model.** Given past slates and choices $s_{0:t-1}, x_{0:t-1}$, we train a network to predict beliefs over the current timestep preferences $u_t$, which we then combine with the current slate $s_t$ to predict a distribution over item choices by using the choice model $P(x_t|u_t, s_t)$: $P(x_t|s_{0:t}, x_{0:t-1}) = \int_{u_t} P(x_t|u_t, s_t)P(u_t|s_{0:t-1}, x_{0:t-1})$. At training time, we supervise this with the actual choice $x_t$ the user made for slate $s_t$ – the network will learn to predict beliefs over preferences which induce behavior which is seen in the training data.

preferences would shift if we exposed them to $\pi'$ *from the beginning*; we call these two the *future* and, respectively, *counterfactual* preference estimation problems, covered respectively in Sec.s 4.1 and 4.2. To facilitate understanding, for each we first describe how to solve it if one had oracle access to the true user internal state dynamics, and then relax this assumption showing how to learn to approximately perform the inference from observable interaction data.

## 4.1 FUTURE PREFERENCES ESTIMATION

We start with the problem of estimating future preferences. That is, for a set of $N$ users, we have access to historical data of their interaction with an RS policy $\pi$, of the form $\mathcal{D}_j = \{s_{0:T}^\pi, x_{0:T}^\pi\}$ for every user $j$. We now are interested in estimating each user's current preferences, and how they would evolve if exposed to a new policy $\pi'$.

**Estimation under known internal state dynamics.** First, we consider the simplified problem in which we have oracle access to the true user internal state dynamics, i.e. $P\big(z_{t+1}|z_t, \pi(s_{0:t}, x_{0:t})\big)$ for any policy $\pi$. Our goal is to estimate the user preferences under the new policy $\pi'$ at a future timestep $H$ – which we denote as $u_H^{\pi'}$. Since we know the user internal state dynamics, via a trivial extension from HMM prediction (Russell & Norvig, 2002), we can use the history of interactions $(s_{0:T}^\pi, x_{0:T}^\pi)$ to estimate the internal state $z_{T+1}$ of the user: $P(z_{T+1}^\pi|s_{0:T}^\pi, x_{0:T}^\pi)$. Then, this filtering estimate can be rolled out with the forward dynamics induced by $\pi'$: $\mathbb{P}(z_H^{\pi'}|s_{0:T}^\pi, x_{0:T}^\pi) = \int_{z_T^\pi} \mathbb{P}(z_H^{\pi'}|z_{T+1}^\pi)\mathbb{P}(z_{T+1}^\pi|s_{0:T}^\pi, x_{0:T}^\pi)$. As preferences $u_H^{\pi'}$ are just one component of the internal state $z_H^{\pi'}$, one can trivially recover a posterior over them.

**Estimation under unknown internal state dynamics.** In practice, one will not have access to an explicit representation of the user's internal state $z_t$, let alone a model of its dynamics. We thus attempt to approximate the NHMM estimation task we are interested in by implicitly learning user preference dynamics from their past interaction data. We train a neural network that, given histories of slates and choices, outputs a belief over preferences $b_{0:T}^\pi(u_t)$. While we do not get direct supervision on preferences, we do get supervision on the choices users make (see Fig. 2). Using the choice model we assume in Sec. 3, we map the belief over preferences, together with the new slate $s_t$, to a distribution over items or choices $b_{0:T}^\pi(x_t)$. At training time, we use the data $\mathcal{D}_j$ for each user and for each time step $t$ to train the model via supervised learning, with a loss defined on the choice distribution outputted by the model, and the actual choice $x_t$ the user made. Such a model learns to perform approximate inference over $u_t$ in the underlying NHMM (see Appendix A).

Then at inference time, to predict future preferences for a user $j$ under a new policy $\pi'$, we sample *simulated user preference trajectories* from the model to approximate Monte Carlo estimation in the NHMM. We use the data $\mathcal{D}_j$ as input to the model to obtain a belief over the next-timestep preferences $u_{T+1}$. We then use such belief and the slate at time $T+1$ from policy $\pi'$, $s_{T+1}^{\pi'} = \pi'(s_{0:T}, x_{0:T})$, to simulate the user's choice $x_{T+1}^{\pi'}$. Treating this extra (simulated) step of interaction history for the user (the slate $s_{T+1}^{\pi'}$ and choice $x_{T+1}^{\pi'}$) as part of the observed history so far, we repeat this process to obtain preference estimates under $\pi'$ for future timesteps. By simulating multiple future trajectories, we obtain beliefs over the expected preferences a user (or a cohort of users) would have at any future timestep. See Algo. 1 for the exact procedure.

While the above method requires the learned future-preference predictor model to generalize to being able to incorporate histories collected under different RS policies than the ones in the training

Figure 3: **Simulating futures.** By iteratively using the future preferences estimation model (Fig. 2) by inputting the observables (shaded, e.g. $x_{0:6}^{\pi}, s_{0:6}^{\pi}$), one can simulate how an existing user's preferences would evolve *in the future* if they interacted with same policy $\pi$ or a different policy $\pi'$ (by selecting future slates with $\pi'$).

data, there is still hope in that real-world datasets of user interaction were been collected under many different deployed policies. This suggests that as long as the slates chosen by $\pi'$ are not too dissimilar from $\pi$ (a mixture policy that comprises all the policies used to gather the data), the network should be able to generalize to predict preference evolutions induced by $\pi'$ too.

## 4.2 COUNTERFACTUAL PREFERENCES ESTIMATION

While the methodology described in Sec. 4.1 is sufficient to estimate the expected preferences and behaviors a user *will have in the future*, it cannot be readily used to estimate *counterfactual* preferences and behaviors: given a set of past interactions $\mathcal{D}_j$ collected under policy $\pi$, we cannot predict what the preferences *would have been for this user* if an alternate policy $\pi'$ had been used from the first timestep of interaction instead of $\pi$ – we denote these *counterfactual preferences* as $u_{0:T}^{\pi'}$. On a high-level, to perform the counterfactual estimation task we want to extract as much information as possible about the user's initial internal state from the historical interactions $\mathcal{D}_j$ we have available for them: even though such interactions were collected with a different policy $\pi$, they will still contain information about the user's initial state (including their initial preferences before interacting with the RS). Then, based on this belief about the user's initial internal state, we want to obtain a user-specific estimate of the effect that another policy $\pi'$ would have had on their preferences.

**Estimation under known internal state dynamics.** Under oracle access to the internal state dynamics, we can first obtain the belief over the initial state of a given user $\mathbb{P}(z_0|s_{0:T}^{\pi}, x_{0:T}^{\pi})$ via NHMM smoothing – which is a trivial extension of HMM smoothing (Russell & Norvig, 2002) – and then roll out the human model forward dynamics with the fixed policy $\pi'$ instead of $\pi$: $\mathbb{P}(z_t^{\pi'}|s_{0:T}^{\pi}, x_{0:T}^{\pi}) = \int_{z_0} \mathbb{P}(z_t^{\pi'}|z_0)\mathbb{P}(z_0|s_{0:T}^{\pi}, x_{0:T}^{\pi})$ – equivalently to Sec. 4.1 when there is no past context but a specific prior for the internal state at $t = 0$.

**Estimation under unknown internal state dynamics.** Without oracle access to the internal state dynamics, again we try to learn to perform this NHMM counterfactual task approximately. One challenge in obtaining supervision for the task is that in our dataset of interactions we never get to see true counterfactuals. We get around this by decomposing the task into two parts, for which we use two separate models: **(1)** estimating the initial internal state for the user (based on the interaction data $x_{0:T}^{\pi}, s_{0:T}^{\pi}$ under $\pi$ which we have available), i.e. train a predictor which approximates the NHMM smoothing distribution $P(u_0|x_{0:T}^{\pi}, s_{0:T}^{\pi})$ which we will denote here as $b_{0:T}^{\pi}(u_0)$ (the belief over initial preferences); and then **(2)** estimating $u_{0:T}^{\pi'}$ conditional on the belief of the initial preferences.

The models for the two tasks are trained with the same method used for predicting future preference estimation (Fig. 2) but with different inputs and supervision signal (Fig. 4). For task **(1)**, the network is trained to predict the *initial* (instead of future) preferences of the user based on later context $x_{1:T}^{\pi}, s_{1:T}^{\pi}$. While the network predicts $P(u_0|x_{1:T}^{\pi}, s_{1:T}^{\pi})$, we can recover the correct smoothing estimate as $P(u_0|x_{0:T}^{\pi}, s_{0:T}^{\pi}) \propto P(x_0^{\pi}|s_0^{\pi}, u_0)P(u_0|x_{1:T}^{\pi}, s_{1:T}^{\pi})$ (see Appendix A).

The network for task **(2)** is obtained by changing the prediction network to also condition on the recovered initial preferences (Fig. 2-4), enabling us to make predictions of the form $P\big(u_{k+1}^{\pi'}|b_{0:T}^{\pi}(u_0), x_{0:k}^{\pi'}, s_{0:k}^{\pi'}\big)$. For training, we first recover initial preference beliefs for each user $j$ in the data $\mathcal{D}_j$ with network for task (1) (which we assume has already been trained). We then train the network for (2) to reconstruct the user $j$'s actual interactions (under $\pi$) simply based on this initial preference estimate (again, similarly to the case in Sec. 4.1 but by additionally conditioning on the initial preference belief). This teaches the network to leverage the information contained user's initial preferences estimate to better estimate their preferences and behaviors based on what their interactions have been so far. In practice, we train these models using a transformer architecture similar to Sun et al. (2019), and we detail in Appendix A why this is a good fit for our tasks.

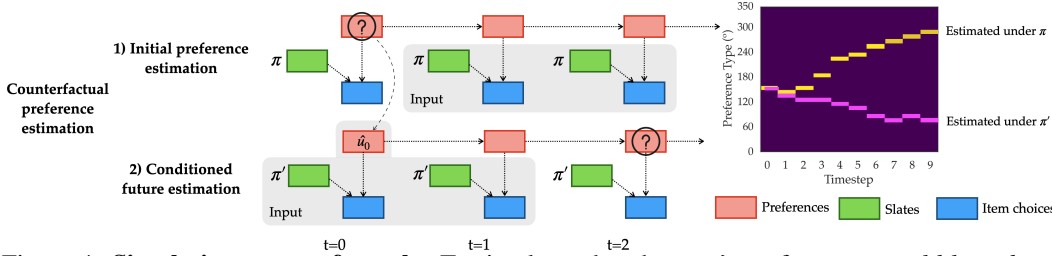

Figure 4: **Simulating counterfactuals.** To simulate what the user's preferences *would have been* under a different policy $\pi'$, we first can use the *initial preference model* to estimate initial preferences based on *later* observables. We can then use the estimated initial preferences to simulate counterfactual preference trajectories for the user under $\pi'$ using a *conditioned* future preference estimation model – where estimates are conditioned on the recovered initial preference belief.

At inference time, we recover an estimate of initial preferences based on interaction data $(s^{\pi'}_{0:T}, x^{\pi'}_{0:T})$ of a new user with a new recommender policy $\pi'$ with model (1), and then estimate the preferences such user would have had under a safe policy $\pi_{\text{safe}}$ with model (2). Such counterfactual preference estimate is obtained with Monte Carlo simulations similarly to Sec. 4.1 with the difference that the network is also conditioned on the initial preferences estimate. See Algo. 2 for the full algorithm and Appendix A for why this approximates the NHMM task.

We now have a way to estimate – for a new policy $\pi'$ – what we would expect its counterfactual impact would have been relative to having deployed $\pi$ (from which we have observational data; note that similarly to future preferences estimation, this procedure too can suffer if $\pi'$ induces a strong distribution shift relative to the training data). We next discuss potential quantitative evaluation metrics defined on these estimates.

## 5 QUANTIFYING UNWANTED SHIFTS AND OPTIMIZING TO AVOID THEM

Given these estimates of preference shifts, it would be useful to have quantitative metrics for whether they are undesirable – as this would enable automatically evaluating policies and even actively optimizing to avoid such shifts. To obviate the difficulty of explicitly defining unwanted shifts, we limit ourselves to defining shifts that we trust ("safe shifts" $u^{\text{safe}}_{0:T}$), and flag preference shifts that largely differ from these safe shifts as potentially unwanted. This is similar to defining a region of the space which we trust *in a risk-averse manner*. The underlying philosophy is that "we don't know what good shifts are, but as long as you stay close to [these shifts], things won't go too poorly". Therefore, we need both a notion of shifts we trust ("safe shifts"), and a notion of closeness (or distance) between induced preference shifts.

**Notation.** We denote as $\hat{r}_t(u^{\text{safe}}_t, \pi)$ the engagement at time $t$ for the content that is chosen *under the policy $\pi$* – if it were evaluated under the safe-shift preferences $u^{\text{safe}}_t$ – i.e. $\hat{r}_t(u^{\text{safe}}_t, \pi) = (x^\pi_t)^T u^{\text{safe}}_t$.

**Distance between shifts.** We choose a metric between shifts such that engagement for the items $x^\pi_{0:T}$ chosen under policy $\pi$ is also high under the preferences one would have had under safe shifts $u^{\text{safe}}_{0:T}$. This is operationalized as $D(u^\pi_{0:T}, u^{\text{safe}}_{0:T}) = \sum_t \mathbb{E}\left[(x^\pi_t)^T u^\pi_t - (x^\pi_t)^T u^{\text{safe}}_t\right] = \sum_t \mathbb{E}\left[\hat{r}_t(u^\pi_t) - \hat{r}_t(u^{\text{safe}}_t, \pi)\right]$ as our notion of distance between $\pi$-induced shifts and safe shifts $u^{\text{safe}}_{0:T}$.

**Safe shifts: $u_0$.** As a first (very crude) proposal for safe shifts, we can consider *no shifts at all*: any deviation from the initial preferences $u_0$ can be flagged as potentially problematic with $D(u^\pi_{0:T}, u_0)$.

**Safe shifts: natural preference shifts (NPS).** Not all preference shifts are unwanted: people routinely change their preferences and behavior "naturally". But what does "naturally" even mean? We propose an idealized notion of natural shifts, by asking how preferences would evolve if the user were "omniscient", i.e. would have access to all content directly and had the ability to process it, unhindered by a small and biased slate offered by an RS. Intuitively, such a user's preferences would still shift over time. Unfortunately this is impractical, as we can never get data from such hypothetical users that can attend to all content when choosing what to consume. As an approximation, we use *random* slates – these slates are still small, but they sampled randomly from the content, eliminating the bias of any RS policy (which in turn can change the user's belief about the distribution of available content). We therefore operationalize "natural preferences" $u^{\pi_{\text{rnd}}}_t$ as the preferences which the user would have interacting with a random RS, and we use $D(u^\pi_{0:T}, u^{\pi_{\text{rnd}}}_{0:T})$ as the metric.

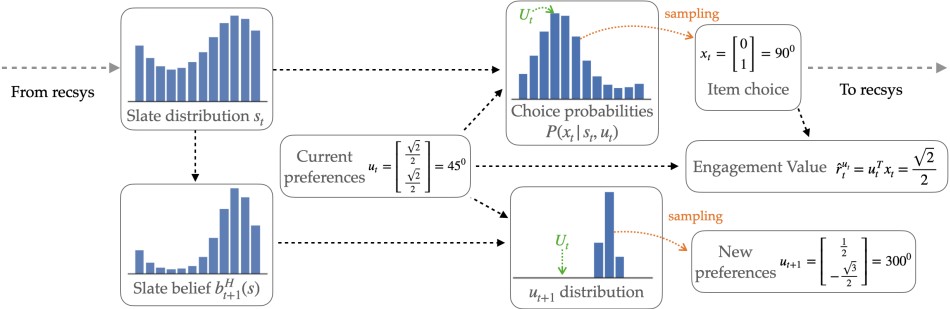

Figure 5: **Ground truth human dynamics.** At each timestep, the user will receive a slate $s_t$. Given the user's preferences $u_t$, the slate $s_t$ induces a distribution over item choices $P(x_t|s_t, u_t)$ from which the user samples an item $x_t$ and receives an engagement value $\hat{r}_t$ (unobserved by the RS). Additionally, $s_t$ induces a belief over the future slates in the user $b_t^H(s)$. In turn $b_t^H(s)$ – together with $u_t$ – induce a distribution over next timestep user preferences, from which $u_{t+1}$ is sampled.

**Penalized training.** By training the models described in Sec. 4, one essentially obtains a human model which can be used to *simulate human choices (and preferences)*. Similarly to previous work, we can use this human model as a simulator for RL training of recommender systems (Chen et al., 2019; Zhao et al., 2019; Bai et al., 2020). The metrics defined above give us a way to augment the reward function to penalize the policy for causing any shifts that we have not explicitly identified as "safe" – in what can be considered a "risk-averse" design choice. We incorporate these metrics in the training of a new RS policy $\pi$ by adding the two distance metrics from above to the basic objective $\sum_t^T \mathbb{E}[\hat{r}_t(u_t^\pi)]$ of maximizing long-term engagement, leading to the updated objective $\sum_t^T \mathbb{E}[\hat{r}_t(u_t^\pi) + \nu_1' \, \hat{r}_t(u_0, \pi) + \nu_2' \, \hat{r}_t(u_t^{\pi_{\mathrm{rnd}}}, \pi)]$ where $\nu_1', \nu_2'$ are hyperparameters – see Appendix B.

# 6 EXPERIMENTAL SETUP

**Why simulation?** To test our method, we need to evaluate both RL policies that interact with users (rendering static datasets of user interaction unsuitable), as well as the metrics themselves, which are defined based on internal preferences (for which we never get ground truth in real interaction). We thus create a testing environment in which we can emulate user behavior and have access to their ground truth preferences. Like previous work (Chaney et al., 2018; Bountouridis et al., 2019; Jiang et al., 2019; Mansoury et al., 2020; Yao et al., 2021), we simulate both a recommendation environment and human behavior. However, unlike such approaches, we use simulated human behavior for testing purposes only: our human model is learned exclusively from data that would be observed in a real-world RS (slates and choices), i.e. that of users (in our case the simulated users) interacting with previous RS policies – meaning our approach could be applied to real user data of this form. A fundamental advantage of testing our methods in a simulated environment is that *we can actually evaluate how well our model is able to recover the preferences of our "ground truth" users, giving us insights about how our methods could perform with real users.*

**Ground truth human dynamics.** See Fig. 9 & 5 for a summary of our environment setup and the ground truth human dynamics we use for testing our method. As mentioned in Sec. 3, we use the conditional logit model as choice model – in our setup, this reduces to $P(x_t = x|s_t, u_t) \propto P(s_t = x)e^{\beta_c x^T u_t}$, with an additional term $P(s_t = x)$ which takes into account how prevalent each item is in the slate (see Appendix C.1) – and we assume this to be also known by our method. We adapt Bernheim et al. (2021) to be our ground truth human preference dynamics. On a high-level, at each timestep users *choose their next-timestep preferences* update their preference to more "convenient" ones – trading off between choosing preferences that they expect will lead them to higher engagement and maintaining engagement under current preferences.

For ease of interpretation, we only consider a cohort of the RS's users whose initial user preferences form a normal distribution around preference $u = 130°$. To showcase preference-manipulation incentives to make users more predictable, we make the choice-stochasticity temperature $\beta_c$ a function of part of preference space one is in, with local optima $\beta_c(80°) = 1$ and $\beta_c(270°) = 4$ (see Appendix C.1 for more details). As users obtain higher engagement value when they act less stochastically, these portions of preference space form attractor points as can be seen in all policies in Fig. 1. While preferences naturally tend to shift towards one of these modes, some RS policies drive preferences to the other mode. As the engagement does not correspond to actual value, converging to the higher local optimum of engagement ($u = 270°$ instead of $u = 80°$) is not necessarily desirable.

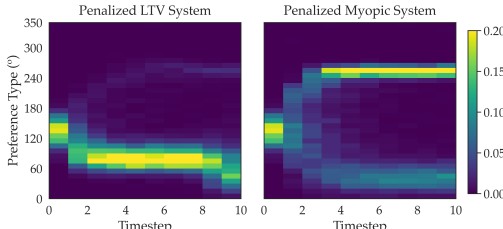
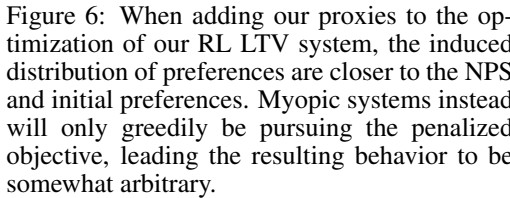
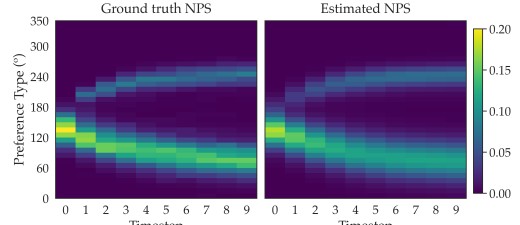

Figure 6: When adding our proxies to the optimization of our RL LTV system, the induced distribution of preferences are closer to the NPS and initial preferences. Myopic systems instead will only greedily be pursuing the penalized objective, leading the resulting behavior to be somewhat arbitrary.

Figure 7: The actual preferences induced by the NPS policy (a random RS $\pi_{\mathrm{rnd}}$) among a cohort of 1000 users (left) vs. a 1000-sampled-trajectory Monte Carlo estimate of what shifts would be induced. Qualitatively, the estimated preferences seem to match the ground truth ones – although slightly more washed out.

**Training human models and penalized RS policies.** For training our human models, we use a BERT model (Sun et al., 2019), and only assume access to a dataset of historical interactions $s_{0:T}, x_{0:T}$. We train myopic and RL RS policies $\pi'$ using PPO (Schulman et al., 2017; Liang et al., 2018) and restrict the action space to 6 possible slate distributions for ease of training. For penalized training we give $\hat{r}_t(u_t^\pi), \hat{r}_t(u_0, \pi'), \hat{r}_t(u_t^{\pi_{\mathrm{rnd}}}, \pi')$ equal weight. See Appendix C for details.

## 7 RESULTS

**Recovering preferences.** Firstly, we confirm that our preference estimation models are able to recover very similar estimates to the equivalent NHMM inference tasks (which, unlike our method, needs access to the ground truth user preference dynamics) – see Appendix D for details, and Fig. 7 for an example of the estimated preferences relative to ground truth ones. This suggests that – under a correct choice model assumption and sufficient similarity between the RS policies represented in the historical datasets, and the ones we are interested in – our method is able to implicitly learn the preference dynamics of a user.

**Hypotheses about metrics.** We now turn to our hypotheses regarding metrics. We hypothesize that our metrics are able to: (**H1**) identify whether policies will induce unwanted preference shifts, and (**H2**) incentivize better behavior when added as a penalty during training.

**Learned human model confound.** With real users, to validate **H1** one would have to rely on approximate computation of the metrics (**estimated evaluation**), as computing the metrics would require using our learned user models of preference dynamics. Additionally, to validate **H2**, one would likely train with simulated interactions from the learned user model (**training in simulation**) – as explained in Sec. 5. The quality of the learned user model would thus be a confound for testing the hypotheses, i.e. the quality of the form of the metrics themselves.

**H1 under oracle dynamics access.** In order to deconfound our experiments from the errors in our estimated human dynamics, we first test these hypotheses assuming oracle access to users – meaning that the policy would be deployed to them – and their dynamics. We first hypothesize that (**H1.1**) by computing the metrics exactly using the preference estimates obtained through NHMM inference (**oracle evaluation**), they are able to flag potentially unwanted preference shifts. We find this to be the case by comparing the oracle evaluation metric values (left of Table 1, "unpenalized") and the actual preference shifts induced by the various RSs we consider (Fig. 1): while unpenalized RL performs better (7.49 vs 5.71) than myopia for engagement $\hat{r}_t(u_t^{\pi'})$, it performs worse with respect to our safe shift metrics. This matches Fig. 1, where RL has more undesired effects.

**H2 under oracle dynamics access.** We additionally hypothesize that (**H2.1**) such metrics (still computed exactly, in **oracle evaluation**) can be used for training penalized LTV systems which avoid the most extreme unwanted preference shifts (and for this training, we will allow ourselves on-policy human interaction data with the ground truth users, as if the RL happened directly in the real world – we call this **oracle training**. For the *penalized* RL RS, the cumulative metric value ("Sum" in Table 1) increases substantially, although it is at the slight expense of instantaneous engagement (Table 1). Qualitatively, we see that the induced preference shifts caused by the RL system seem closer to "safe shifts" (Fig. 6), supporting **H1.1** in that high metric values qualitatively match shifts that seem more desirable. Overall, we see that with oracle access, the metrics capture

what we see as qualitatively undesired shifts, and that optimizing for them directly (penalized RL) produces policies with slightly lower engagement, but drastically better at avoiding such shifts.

**H1 and H2 with learned user dynamics.** Our findings support our hypotheses under oracle access. However, in real life we will not have access to the underlying dynamics model, or even the ability to interact on-policy with users as we train RL policies, due to the high cost and risk of negative side effects for collecting data with unsafe policies. Therefore, we wish to show that even with *estimated metrics and simulated interactions* (based on the learned models), (**H1.2**) **estimated evaluation** and (**H2.2**) **training in simulation** (described above) are still able to respectively flag unwanted shifts and penalize manipulative RL behaviors. Table 2 shows that – although the estimated metrics can differ from the ground truth ones somewhat substantially (see Oracle Eval. at the top vs Estimated Eval. at the bottom) – importantly the relative ordering of the policies ranked by our estimated values stay the same: the penalized RL policy trained in simulation actually has (under oracle evaluation) higher cumulative reward than the unpendalized one, and the estimated evaluation keeps that ranking (even though is more optimistic about the unpenalized reward).

Table 1: **Results under oracle training and evaluation.** Both here an in Table 2 we report the cumulative expected engagement under a variety of different preferences (as explained in Sec. 5). The values are averaged across 1000 trajectories (standard errors are all $< 0.1$). By looking at the safe shift metrics $\hat{r}(u_0)$ and $\hat{r}(u_t^{\text{rnd}})$, we see that penalized systems stay significantly closer to safe shifts than unpenalized ones.

Table 2: **Effect of estimating evaluations and simulating training for LTV.** We see that the estimated evaluations of trained systems strongly correlate with the oracle evaluations (and importantly, maintain their relative orderings). A similar effect occurs when training in simulation rather than by training with on-policy data collected from real users.

**Table 1**

| | | Oracle Training | | |
|---|---|---|---|---|
| | | *Unpenalized* | | *Penalized* | |
| | | Myopic | RL | Myopic | RL |
| **Oracle Eval** | $\hat{r}_t(u_t^{\pi'})$ | 5.71 | 7.49 | 6.20 | 5.28 |
| | $\hat{r}_t(u_0, \pi')$ | 1.99 | -0.08 | 3.61 | 6.21 |
| | $\hat{r}_t(u_t^{\pi\text{rnd}}, \pi')$ | 2.01 | -1.09 | 3.10 | 4.57 |
| | Sum | 9.69 | 6.33 | 12.90 | 16.05 |

**Table 2**

| | | Oracle Training | | Training in Sim. | |
|---|---|---|---|---|---|
| | | *Unpen.* | *Penal.* | *Unpen.* | *Penal.* |
| **Oracle Eval** | $\hat{r}_t(u_t^{\pi'})$ | 7.49 | 5.28 | 6.40 | 5.48 |
| | $\hat{r}_t(u_0, \pi')$ | -0.08 | 6.21 | -1.24 | 5.61 |
| | $\hat{r}_t(u_t^{\pi\text{rnd}}, \pi')$ | -1.09 | 4.57 | -1.83 | 4.43 |
| | Sum | 6.33 | 16.05 | 3.36 | 15.52 |
| **Est. Eval** | $\hat{r}_t(u_t^{\pi'})$ | 5.58 | 5.42 | 6.49 | 5.78 |
| | $\hat{r}_t(u_0, \pi')$ | 1.28 | 5.57 | -0.80 | 4.94 |
| | $\hat{r}_t(u_t^{\pi\text{rnd}}, \pi')$ | 2.05 | 3.88 | 1.48 | 4.41 |
| | Sum | 8.91 | 14.87 | 7.17 | 15.15 |

## 8 DISCUSSION

**Summary.** In conclusion, our contributions are 1) proposing a methodology to estimate the preference shifts which would be induced by recommender system policies before deployment, and 2) providing a formalism for defining distance metrics from safe preference shifts, which can be used both for evaluation and to train RSs which penalize unwanted shifts. As dynamics of preference are learned (rather than handcrafted), our method could be applied to real user data and has the potential for higher accuracy than handcrafted models due to massive amounts of behavior data. Further, while there is no ground truth for human preferences, verifying the model's ability to anticipate behavior can give confidence in using it to evaluate and penalize undesired preference shifts.

**Limitations and future work.** We conducted our evaluation in simulation, which was necessary for testing the metrics themselves – for this we required ground truth access to user preferences. But since we used a specific dynamics model in our simulator, we cannot guarantee that our results translate when the method is applied to real users with real preference dynamics. We expect real users to have more complex dynamics, with preference changes mediated by beliefs outside of content distribution as in our simulator, such as beliefs about the world; while our method makes no assumptions about the structure of this internal space, it might require a lot more (and more diverse) data to capture these effects. Further, we also limited our evaluation to a setting explicitly designed such that an RL system would have incentives to manipulate preferences – not all real settings will necessarily have that property. Moreover, crucially our method requires assuming access to the user choice model: one could obviate this by predicting behavior and defining metrics on behavior, but this would mean losing the latent preference structure. Additionally, even what we call "preferences" – de-noised behavior – is limiting as it doesn't lend itself to capturing long-term preferences.

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

## A  PREFERENCE ESTIMATION

**Future preference estimation with Algorithm 1 and why it approximates NHMM prediction.**

Algorithm 1 is used to obtain the belief over the preferences of a user would have at timestep $H$ – assuming that the user interacted for $T$ timesteps with a policy $\pi$ (for which we have actual interaction data), and then interacted from timestep $T$ to $H$ with a policy $\pi'$. We refer to this belief as $b_{0:T}^{\pi}(u_H) = P(u_H^{\pi'}|s_{0:T}^{\pi}, x_{0:T}^{\pi})$.

The first model inference pass in Algorithm 1 resulting in $b_{0:T}(u_{T+1})$ will be a distribution over $u_{T+1}$ conditioned on $s_{0:T}^{\pi}, x_{0:T}^{\pi}$. A minimizer for this prediction problem would be for the network to represent the actual belief that would be obtained by performing the prediction step in the NHMM (described in Sec 4.1). This is because the expected cross-entropy between user choices and the induced choice distribution (induced by the preference belief prediction) would be minimized.

One limitation is that there are possibly multiple beliefs over preferences that induce the same exact distribution over choices: in that sense, the correct preference belief might be unidentifiable. Experimentally, we don't find this to be a problem: the choice (see the "Preference estimation model form" heading of this Appendix) of having the representation of the preference belief be a mixture of Von Mises distributions – which can be thought of as Normals over a circle – is already a good enough inductive bias to be able to recover good preference beliefs (see Appendix D).

In the NHMM, note that the unrolling the forward model (i.e. computing $\mathbb{P}(u_H^{\pi'}|z_{T+1}^{\pi})$) could be computed exactly, or be performed with Monte Carlo estimation, by sampling many internal states $z_{T+1}^{\pi}$ from the forward prediction distribution, and then sampling internal state evolutions according to the dynamics. In our algorithm, by sampling choices and slates then conditioning on them, we are approximating the Monte Carlo estimation approach. When simulating each user choice, the model can be thought of as implicitly sampling a preference for this simulated user (from the preference belief) and then sampling a choice from the corresponding choice distribution. This means that each simulation rollout should be equivalent to "sampling a user" (according to the distribution of users in the data) and then sampling their choices.

---

**Algorithm 1:** Predicting future user preferences at timestep $H$ under policy $\pi'$

---

**Inputs:** past interactions $x_{0:T}^{\pi}$, $s_{0:T}^{\pi}$, a RS policy $\pi'$ with signature $(s_{0:k}, x_{0:k}) \to s_{k+1}$, a future
   preference estimator $\hat{P}_f$ with signature $(s_{0:k}, x_{0:k}) \to (b_{0:k}(u_{k+1}), b_{0:k}(x_{k+1}))$, a horizon $H$,
   number of Monte Carlo simulations $N$;

**if** $x_{0:T} = s_{0:T} = \emptyset$ **then** // if no past interaction data is given
    | Sample slate $s_0 \sim \pi'(\emptyset, \emptyset)$ and imagine user choice $x_0 \sim \hat{P}(x_0|\emptyset, \emptyset)$ ;    // we now have past
    | interactions with $T = 0$

**for** $i = 0$*; $i < N$; $i + +$ **do** // sample N simulated trajectories
    | **while** $k = T$*; $k < H$; $k + +$ **do**
    |   | $b_{0:k}(u_{k+1}), b_{0:k}(x_{k+1}) = \hat{P}_f(s_{0:k}, x_{0:k})$ ;            // estimate pref. and choices
    |   | $s_{k+1} \sim \pi'(x_{0:k}, s_{0:k})$ ;                           // sample next timestep slate
    |   | $x_{k+1} \sim b_{0:k}(x_{k+1})$ ;                             // simulate a user choice
    |   | Add $x_{k+1}$ and $s_{k+1}$ to the current simulated trajectory's history;

Average $b_{0:k-1}(u_k)$ and $b_{0:k-1}(x_k)$ across the $N$ futures (for each $k$, with $T < k \le H$);
**Result:** Belief over future pref. $b_{0:T}^{\pi}(u_k)$ and behaviors $b_{0:T}^{\pi}(x_k)$ for each $k$ s.t. $T < k \le H$.

---

**Initial preference prediction model output correction**

Below, we show that one can recover the smoothing estimate $P(u_0|x_{0:t}, s_{0:t})$ from the predicted preferences $P(u_0|x_{1:t}, s_{1:t})$ which will be a biased estimate of the initial preferences (as it does not incorporate the information from timestep $t = 0$) . Note that:

$$P(u_0|x_{0:t}, s_{0:t}) = \frac{P(u_0|x_0, s_0)P(x_{1:t}, s_{1:t}|u_0)}{P(x_{0:t}, s_{0:t})} = \frac{P(u_0|x_0, s_0)}{P(x_{0:t}, s_{0:t})} \frac{P(u_0|x_{1:t}, s_{1:t})P(x_{1:t}, s_{1:t})}{P(u_0)} \tag{1}$$

$$= \frac{P(x_0, s_0|u_0)P(u_0|x_{1:t}, s_{1:t})P(x_{1:t}, s_{1:t})}{P(x_{0:t}, s_{0:t})P(x_0, s_0)} \propto P(x_0, s_0|u_0)P(u_0|x_{1:t}, s_{1:t}) \tag{2}$$

The first equality is given by the definition of smoothing applied to $t = 0$ (Russell & Norvig, 2002). The second equality is obtained by using Bayes Rule on the backwards message $P(x_{1:t}, s_{1:t}|u_0)$, and the third is obtained by using Bayes Rule on $P(u_0|x_0, s_0)$. Finally, we can ignore $P(x_{1:t}, s_{1:t})$, $P(x_{0:t}, s_{0:t})$, and $P(x_0, s_0)$ as they are constants.

**Future preference estimation with Algorithm 2 and why it approximates NHMM prediction.**

A similar argument to one in the above section can be made as to why such a initial preference estimation network would approximate the corresponding NHMM smoothing task.

However, for the second step of counterfactual preference estimation, one issue arises. Relative to having access to the full dynamics of the internal state, when performing approximate inference with our model we lose some information: we are only able to recover a belief over the initial *preferences*, whereas the NHMM with full dynamics access would be able to recover a belief over the *full internal state* of the user. This will reduce the accuracy of our counterfactual estimation, but is the best we can do without further assumptions.

Mathematically, we approximate the NHMM target distribution $P(u_T^{\pi'}|x_{0:t}^\pi, s_{0:t}^\pi)$ as:

$$P(u_T^{\pi'}|x_{0:t}^\pi, s_{0:t}^\pi) \approx \int_{u_0} P(u_T^{\pi'}|u_0)P(u_0|x_{0:t}^\pi, s_{0:t}^\pi) = P\big(u_T^{\pi'}|b_{0:t}^\pi(u_0)\big) \tag{3}$$

$$= \sum_{x_{0:T-1}^{\pi'}, s_{0:T-1}^{\pi'}} P(x_{0:T}^{\pi'}, s_{0:T}^{\pi'})P\big(u_T^{\pi'}|b_{0:t}^\pi(u_0), x_{0:T}^{\pi'}, s_{0:T}^{\pi'}\big) \tag{4}$$

where we the last expression is approximated with a Monte Carlo estimate detailed in Algorithm 2[1] (similarly to what was done in Algorithm 1). To do well at this second trajectory reconstruction task, the network necessarily needs to learn how to make best use of the belief over initial preferences, and implicitly learn their dynamics, as for the future preference estimation task.

---

**Algorithm 2:** Predicting counterfactual user preferences at timestep $T$ under policy $\pi'$, given $t$ timesteps of interaction data with $\pi$.

---

**Inputs:** past interactions $x_{0:t}^\pi, s_{0:t}^\pi$, a RS policy $\pi'$ with signature $(s_{0:k-1}, x_{0:k-1}) \rightarrow s_k$, an initial preference estimator $\hat{P}_i$ with signature $(s_{0:k-1}, x_{0:k-1}) \rightarrow b_{0:k}(u_0)$, a conditioned future estimator $\hat{P}_c$ with signature $(s_{0:k-1}, x_{0:k-1}, b_{0:k}(u_0)) \rightarrow (b_{0:k}(u_{k+1}), b_{0:k}(x_{k+1}))$, a horizon $T$, a constant $N$;

$b_{0:t}^\pi(u_0) = \hat{P}_i(s_{0:t}^\pi, x_{0:t}^\pi)$ ;                 // initial pref. belief given interactions with $\pi$

$\hat{P}_f = \hat{P}_c(b = b_{0:t}^\pi(u_0))$ ;                 // future pref predictor conditioned on init belief

$\big(b_{0:k-1}^\pi(u_k^{\pi'}), b_{0:k-1}^\pi(x_k^{\pi'})\big)_{0 < k \leq T} =$ Algorithm 1 $\Big(\emptyset, \emptyset, \pi', \hat{P}_f, T, N\Big)$

**Result:** Distributions of counterfactual preferences and behaviors under policy $\pi'$

---

**Preference estimation model form**

We have three models for preference estimation: respectively initial, counterfactual, and future preference estimators $\hat{P}_i, \hat{P}_c, \hat{P}_f$. Any sequence model, such as RNNs or transformers, would be appropriate for these tasks that have variable number of inputs.

One detail of note that was omitted from Fig. 2 is that – to enable to represent multi-modal beliefs over preference space – we let the models' output be parameters of multiple Von Mises distributions and additionally some weights $w$. The $w$ weighted average of these distributions will form the predicted belief over $u_t$.

In architecture, we use a similar form to that of BERT4Rec (Sun et al., 2019) for ease of performing inference on future or past preferences given contexts of interaction, as described in Fig. 8.

---

[1]This algorithm is not actually used in it's pure form in the experiments. Our choice of "safe policy" $\pi_{\text{rnd}}$ happens to choose constant slates (i.e. a uniform distribution no matter the history), so there is a shortcut to the procedure: by simply setting the inputted slated for counterfactual prediction to be uniforms, and predicting preferences without any user choices, one can train the model to directly output the belief over counterfactual preferences for any timestep, across the whole userbase.

Using transformers is not new to the context of recommendation for click prediction (Sun et al., 2019), but to our knowledge we are the first to leverage the flexibility of conditioning schemes enabled by the BERT architecture (Devlin et al., 2019; Joshi et al., 2020; Wang & Cho, 2019).

Specifically, we train 3 transformers (one for prediction, one for smoothing, and one for counterfactual estimation), although preliminary results show that one can train a single transformer to perform all three tasks without a significant loss in performance. All together, these 3 transformers constitute the "learned human model" which we refer to throughout the text.

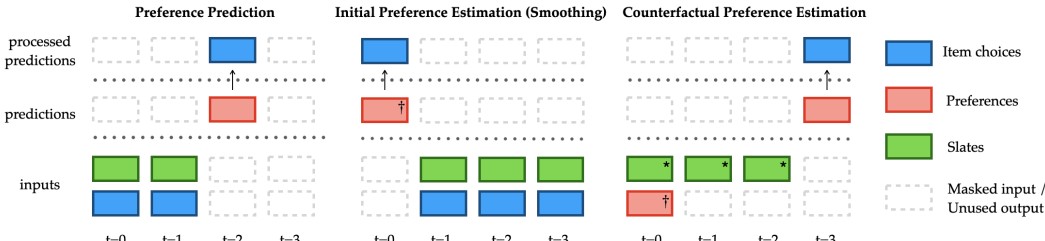

Figure 8: **BERT representation of the inference tasks.** While our method is compatible with any sequence model, we choose to use a BERT transformer models. **Left:** estimation of the user's future preferences and choice at t=2, given the interaction history history so far (this modality of prediction is closest in setup to (Sun et al., 2019)). **Middle:** recovering a belief over the initial preferences and choice of the user based on later interactions. **Right:** conditioning on the estimate of initial preferences (†) recovered from the smoothing network one can estimate counterfactual preferences and choices under slates (*) (chosen from a policy $\pi'$ we are interested in) and imagined choices (not shown due to space).

## B   PENALIZED RL

**The underlying MDP**

One could cast the recommendation problem as a POMDP (Lu & Yang, 2016; Mladenov et al., 2019) in which the state of the environment is hidden and contains the user's internal state, which evolves over time. Equivalently, one can consider the belief-MDP induced by the recommender POMDP (Kaelbling et al., 1998), and approximate a solution to such belief-MDP via Deep-RL with a policy trained with observation histories as input (this is theoretically sufficient for the policy to recover a belief over the current hidden state and take the optimal action). The action space will be given by the space of possible slates that the RS can choose. The reward signal will be the expected reward for the current timestep $\mathbb{E}\big[\hat{r}_t(u_t^\pi)\big]$ (or with the extra terms for the proxies in the case of penalized training). The introduction of expectation can be thought of as expected SARSA (Sutton & Barto, 1998), as argued in (Ie et al., 2019).

**Penalized RL training**

The full set of steps to run RL training are as follows: once the human models described in Sec. 4 are trained, one can use them to simulate user trajectories and compute penalty metrics for such trajectories (see Algorithm 3). One can then optimize the RL policy based on the on-policy simulated trajectory rollouts.

**Reduction of distances to final penalized objective**

Note that the full penalized RL objective $\left( \sum_t^T \mathbb{E}\big[\hat{r}_t(u_t^\pi)\big] \right) - \nu_1 D(u_{0:T}^\pi, u_0) - \nu_2 D(u_{0:T}^\pi, u_{0:T}^{\pi_{\mathrm{rnd}}})$

for our choice of distance function $D$ reduces to $\sum_t^T \mathbb{E}\big[\hat{r}_t(u_t^\pi) + \nu_1' \, \hat{r}_t(u_0, \pi) + \nu_2' \, \hat{r}_t(u_t^{\pi_{\mathrm{rnd}}}, \pi)\big]$ for some choice of $\nu_1', \nu_2'$ which can be treated as hyperparameters of how much we want to value the engagement under each safe policy preferences relative to the engagement under the main policy.

## C   EXPERIMENT DETAILS

### C.1   GROUND TRUTH USERS

**Reduction of logit model to our case.**

---

**Algorithm 3:** Generating a trajectory for RL training and computing metrics

---

**Inputs:** Initial, counterfactual, and future preference estimators $\hat{P}_i, \hat{P}_c, \hat{P}_f$; a policy $\pi$, a safe policy $\pi_{\text{safe}}$, a horizon $H$, a constant $N$.

Sample slate $s_0^\pi \sim \pi(\emptyset, \emptyset)$ and imagine user choice $x_0^\pi \sim \hat{P}_i(x_0|\emptyset, \emptyset)$;

**for** $t = 1;\ t \le H;\ t++$ **do**

$\quad b_{0:t-1}^\pi(u_0) = \hat{P}_i(s_{0:t-1}^\pi, x_{0:t-1}^\pi)$ ;                    // current belief over initial preferences

$\quad b_{0:t-1}^\pi(u_t), b_{0:t-1}^\pi(x_t) = \hat{P}_f(s_{0:t-1}^\pi, x_{0:t-1}^\pi)$ ;                    // belief over pref and choices

$\quad \mathbb{E}[b_{0:t-1}^\pi(u_t^{\pi_{\text{safe}}})] \in \text{Algorithm 2}\big(t, \pi_{\text{safe}}, \hat{P}_c, \hat{P}_i, x_{0:t-1}^\pi, s_{0:t-1}^\pi\big)$ ;      // belief over counterfactual

$\quad$ preferences under safe policy for this user

$\quad s_t^\pi \sim \pi(x_{0:t-1}^\pi, s_{0:t-1}^\pi)$ ;                    // sample slate

$\quad x_t^\pi \sim b_{0:t-1}^\pi(x_t)$;                    // imagine a user choice

$\quad D_t(u_t^\pi, u_t^{\text{safe}}) = \mathbb{E}_{u_t^\pi, u_t^{\pi_{\text{safe}}}, x_t^\pi}\big[(x_t^\pi)^T u_t^\pi - (x_t^\pi)^T u_t^{\pi_{\text{safe}}}\big]$ ;      // compute penalty metric(s) for

$\quad$ timestep

$\quad r_t^{RL} = \mathbb{E}[r_t] + D_t(u_t^\pi, u_t^{\text{safe}})$

**Return:** User-RS interactions and training rewards for the simulated trajectory;

---

In our experimental setup, the traditional conditional logit model $P(x_t = x|s_t, u_t) = \frac{e^{\beta_c x^T u_t}}{\sum_{x \in s} e^{\beta_c x^T u_t}}$ doesn't apply directly in this form, as we consider slates to be distributions rather than sets of discrete items. Intuitively, user's choices should still depend on the slate: the proportion of a certain item in the current slate (one can think of this as the proportion of a certain item *type*), should influence the probability of the user of selecting that item (type). We operationalize this as $P(x_t = x|s_t, u_t) \propto P(s_t = x)e^{\beta_c x^T u_t}$, with an additional term $P(s_t = x)$ which takes into account the proportion of each item (type). Note that this also mathematically corresponds to the generalization of the traditional logit model: when the slate is discrete, $P(s_t = x)$ will simply be an indicator for whether the item is in the slate, leading to the traditional logit model form.

**Feed belief update**

Our ground truth user has a belief over future slates $b_t^H(s)$. Users' initial belief matches the content feature distribution itself $b_0^H(s) = \mathcal{D}$. After receiving a slate $s_t$, the user's induces a belief $b_t^H(s) \propto s_t^3$ over the future slates in the user, i.e. the user will expect the next feeds to look like the most common items in the current feed, as a result of availability bias (MacLeod & Campbell, 1992).

**Lack of no-op choice.**

While the assumption that user must pick item from the slate is unrealistic, this could be resolved by adding an extra no-op choice to every slate (Sunehag et al., 2015).

**Prefernce shift model.**

Our preference shift model is inspired by Bernheim et al. (2021), but adapted to our experimental setup as described below. The choice of preferences is modulated by a "mindset flexibility" parameter $\lambda$ which captures how open they are to modifying their current preferences. Users assign value to the choice of next-timestep preferences $u_{t+1}$ as: $V\big(u_t, b_t^H(s), u_{t+1}, \lambda\big) = \mathbb{E}_{x_{t+1} \sim b_t^H(s)}[\lambda \hat{r}_{t+1}(u_t) + (1-\lambda)\hat{r}_{t+1}(u_{t+1})]$, where with $\hat{r}_{t+1}(u_t)$ indicates the engagement value obtained by the user under the choice $x_{t+1}$ and preference $u_t$. Users pick their next timestep preferences also according to the conditional logit model, but over their expected value of such preferences choices: $P(u_{t+1}|b_t^H(s), u_t, \lambda) \propto e^{\beta_d V(u_t, b_t^H(s), u_{t+1}, \lambda)}$. Intuitively, users update their preference to more "convenient" ones – ones that they expect will lead them to higher engagement value. The main change we introduce from the original model is incorporating the belief over future feeds.

User's preference-flexibility parameter is given by $\lambda = 0.9$, and their initial preferences are drawn from a normal distribution[2] around preference $u = 130°$ and standard deviation $20°$.

---

[2] Technically one should use Von Mises distributions – a distribution similar to normals, but for which the domain is a circle. As Von Mises distributions are not implemented in numpy (Harris et al., 2020), for simplicity we use clipped normal distributions (disregarding probability mass beyond $180°$ in either direction) in all places except for the the transformer output (which was implemented in PyTorch (Paszke et al., 2019), which has a Von Mises implementation)

## C.2 LEARNED HUMAN MODELS

For our learned human models, we use BERT transformers with 2 layers, 2 attention heads, 4 sets of Von Mises distribution parameters, a learning rate of $0.00003$, batch size of 500, and 100 epochs. We train on the data described below.

## C.3 SIMULATED DATASET

See Fig. 9 for a summary of how content is generally instantiated in our setup.

We set the distribution of content in such a way that it forms a uniform distribution across features, that is $\mathcal{D} = \text{Uniform}(0°, 360°)$. We simulate historical user interaction data with a mixed policy $\pi$ which is similar (but not equal to) a random policy in half the rollouts and for the other half is goal-directed:

- Half of the data is created with a RS policy which chooses an action uniformly among a set of possible slates (distributions over the feature space with means $0°, 10°, ..., 340°, 350°$, and standard deviations equal to $30°$ or $60°$). Each of these slate types can be thought of as a slate which contain mostly one specific type of content.
- The other half of the data is created with a RS which chooses $s_t = \mathcal{D} = \text{Uniform}(0°, 360°)$ 80% of the time (i.e. the slate that would be chosen by the random RS $\pi_{\text{rnd}}$), and chooses a random action from the same set of possible slates the remaining 20% of the time.

This is to simulate the setting in which the safe policy we are interested in (the random NPS policy) is similar to previously deployed ones that are represented in the data (although not the same, so the network still has to generalize across RS policies at test time).

Our dataset consists of $100k$ user trajectories (slates and observations), with length of each trajectory $T = 10$.

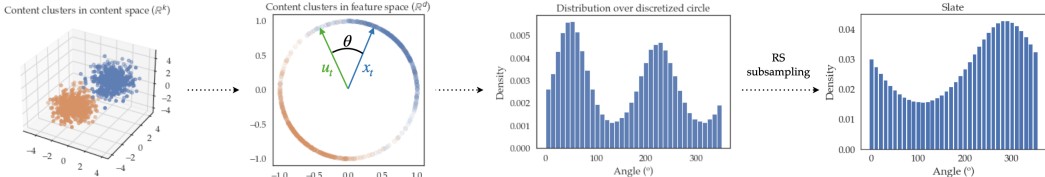

Figure 9: **a)** $\to$ **b)** The content can be mapped to an empirical distribution over feature space $\mathcal{D}$ in $\mathbb{R}^d$. We consider dimension $d = 2$ for ease of visualization. Restricting preferences and choices to be unit vectors, one can think of them as a points on a circle: the engagement value $\hat{r}_t$ will thus be related to the angle $\theta$ between $u_t$ and $x_t$. **c)** We discretize this circular preference and feature space into $n = 36$ bins (i.e. binning the angles) which enables to visualize distributions over preferences and over content features as histograms over angles. **d)** We model slates $s_t$ as categorical distributions over the discretized $n$-bin feature space.

## C.4 RS TRAINING

For RL optimization, we use PPO (Schulman et al., 2017) trained with Rllib (Liang et al., 2018). The action space of the recommender system is given by distributions over feature space with means $0°, 60°, ..., 260°, 320°$, and standard deviations equal to $60°$. As observations to the system, we provide the current slate, the previous user choice, and the current estimates from the HMM for smoothing, filtering, and natural preference shift counterfactual distributions, in order to increase training speed (note that these will not change the optimal policy). All policies are recurrent so they are able to reason about the history of interactions so far.

We use batch size 1200, minibatch size 600, 4 parallel workers, $0.005$ learning rate, 50 gradient updates per minibatch per iteration, policy function clipping parameter of $0.5$, value function clipping parameter of 50 and loss coefficient of 8, with an LSTM network with 64 cell size. Training runs in less than 30 minutes for each condition on a MacBook Pro 16" (2020).

## D RESULTS

**Quality of preference estimation**

We baseline the quality of our counterfactual estimates of user preferences and behaviors relative to the performance of NHMM with full access to the human dynamics – as that is the best performance we could possibly hope for. The metrics we use to evaluate the performance of the transformer relative to the HMM are the 1) *relative prediction loss* (RPL) as the ratio between the transformer loss and the HMM loss ($\frac{\mathcal{L}_{Tr}}{\mathcal{L}_{HMM}}$), and the 2) *relative prediction accuracy* (RPA), defined similarly as $\frac{Acc_{Tr}}{Acc_{HMM}}$. Given that NHMM is the gold standard of estimation (as we are doing exact inference), RPL should be larger than 1 and RPA should be lower than 1. The closer to 1 the numbers are, the better performing our approximate inference methods are. For all the models, we train on a 75k user interaction trajectories dataset obtained from the distribution described in C. For all the tasks below, we use the methodology developed in Sec. 4.1.

**Future preference estimation** We train a transformer model as in the left portion of Fig. 8, by predicting – for any given timestep $t$ – the preferences $u_t$ (and thus also a distribution over item choices) based on slates $s_0, \ldots, s_{t-1}$ and the previous observations $x_0, \ldots, x_{t-1}$. All other inputs are masked. We evaluate on a held out dataset of 25k user interaction trajectories, which are obtained from the same distribution as the training data. The preference RPL and RPA (averaged across all timesteps) are $1.08$ and $0.88$.

**Initial preference estimation** We train a transformer model as in the middle portion of Fig. 8, by predicting the preferences $u_0$ (and thus also a distribution over item choices) based on slates $s_1, \ldots, s_T$ and the observations $x_1, \ldots, x_T$. All other inputs are masked. We evaluate on the same held out dataset as the prediction case. The preference RPL for the initial timestep is $1.08$, and the preference RPA is $0.81$.

**Counterfactual preference estimation** We train a transformer model as in the right portion of Fig. 8, by predicting – for any given timestep $t$ – the preferences $u_t$ (and thus also a distribution over item choices) based on slates $s_0, \ldots, s_T$ and the observations $x_1, \ldots, x_T$ – i.e. reconstructing the original trajectory from the smoothing estimate obtained from the network described above. We evaluate on a held out dataset of 25k user interaction trajectories obtained *from the natural preference shift policy* $\pi_{\text{rnd}}$ – showcasing the generalization performance of our method to different policies. The preference RPL (averaged across timesteps) is $1.16$, and the preference RPA is $0.63$ – showing that although the task is more challenging for the models, performance doesn't degrade drastically relative to the NHMM. Fig. 7 shows this qualitatively: the counterfactual network is able to estimate what natural preference shifts would look like relatively well relative to the ground truth across the population.

