# OpenReview forum: "Estimating and Penalizing Induced Preference Shifts in Recommender Systems"
_ICLR.cc/2022/Conference — ICLR 2022 Submitted_

### Official Review · Reviewer_HPbv · 2021-11-01

**Correctness:** 4
**Technical Novelty And Significance:** 3
**Empirical Novelty And Significance:** 3
**Recommendation:** 6
**Confidence:** 3

**Details Of Ethics Concerns:**

The paper aims to solve ethical problems. BUT, it does not mean it could not induce other problems in the process.

**Main Review:**

Reasons to accept:
*The ideas presented in this paper are somewhat taboo, and commercial companies are not likely to do this type of research. This is another reason to encourage studying it.

*The methods the authors suggest are practical, and their results lay the foundations for “neutral” recommendation policies.

Reasons to reject:

*The results rely on modeling assumptions, for instance, their choice modeling. Different assumptions can change the conclusions entirely.

*The work has many limitations (despite that the authors address most of them). For instance, I am not sure that random recommendation is a good benchmark to test for preference shifts; the authors did not motivate this selection well enough.

I think the topic of this paper should be a core one at the ethical ML research direction, and yet it does not receive enough attention (especially not from industry). The counterfactual arguments and tools the authors propose are essential for conducting a what-if analysis like the one they propose, so the behavioral assumptions are somewhat inevitable. Overall, I think the community can benefit from the ideas presented in the paper, so I vote for (weakly) acceptance.

There are several typos in the paper; here are some of them:
-page 2: the \tilde sign is presented incorrectly with the percentages.
-page 2: "we e active", e -> take?
-page 18 "the the" inside the footnote.

**Summary Of The Paper:**

The authors address the challenge of preference shifts, a potentially undesired effect of recommendation systems (RSs). A preference shift is a byproduct of any RL system, in which the systems drive users to places (i.e., in their particular preference representation) where they are easily satisfiable. This can happen, e.g., by positive feedback loops. The main challenge of tacking this undesired effect is that we lack the “natural preferences” or a robust way to predict preference shift given a recommendation policy. The authors suggest building a system, which is partially data-driven and partially based on behavioral assumptions to simulate recommendation policies and find safe shifts --- shifts ensuring that users’ preferences will not be influenced too much by the specific policy. They explain the ingredients of the system and experimentally test it.

**Summary Of The Review:**

The paper addresses a real-world problem and provides tools to deal with it. I vote for weak acceptance.

---

> ### Author Response · Authors · 2021-11-14
> **Response to Reviewer HPbv**
>
> Thank you so much for your thoughtful review, and recognizing how important and understudied our problem setting is. We also think that this is an important direction which should be further explored in ethical ML, especially in the context of recommender systems.
>
> While your review was almost entirely positive, we were surprised as to your evaluation. Elaborating further on what could be improved would be very helpful for us!
>
> Below we attempt to address the specific limitations which you highlighted in your review:
>
> ---
>
> > The results rely on modeling assumptions, for instance, their choice modeling. Different assumptions can change the conclusions entirely.
>
> While we acknowledge that our method does require users to choose content in ways that are similar to the assumed choice model, we don’t think this could change the conclusions of our study _entirely_: we would still be showing that if the choice model is correctly specified, our method can be used successfully to recover user preferences (and these estimates can then be used downstream to estimate and penalize effects of policies). However, we realize that correctly specifying the choice model is a challenging problem.
>
> That being said, an important direction of future research would be to see the degree to which our method is robust to recovering estimates when there is mis-specification in the choice model, and investigating ways to estimate the form of the user choice model.
>
> ---
>
> > I am not sure that random recommendation is a good benchmark to test for preference shifts; the authors did not motivate this selection well enough.
>
> Unfortunately, we had to cut some of the motivation for our approach to Natural Preference Shifts from the main text because of space constraints. Below we attempt to propose another perspective on the motivation proposed in Section 5 (under “Safe shifts: natural preference shifts”)
>
> The high level idea is as follows: given the difficulty of specifying what “unwanted preference shifts” are directly, we propose to compare the shifts induced by the system to the shifts the user would have had in an idealized setting, i.e. what would have been the user’s preference evolution if they weren’t affected by the recommender policy’s influence, and if they were changing their preferences in an ideal way (based on sound judgements and beliefs, not affected by temptation, etc.)? Clearly, this notion of “natural preference shifts” is not easily operationalizable, leaving us with the question of how we can approximate this notion in practice.
>
> Given that simulating users that have perfect beliefs and judgement is clearly extremely challenging, we instead try to approximate this ideal notion by focusing on removing as much recommender system influence as possible – and choose a random recommender as it is the most non-agentic recommender we could think of. This is clearly not a perfect choice, but we think it can provide an interesting starting point.
>
> In summary, it’s useful to differentiate between the _idea of using natural shifts_, and _our specific practical operationalization of it_ which uses random slates. We would like to note that we are not married to our specific choice of the latter.
>
> ---
>
> We would also like to thank you for taking the time to tell us the typos you found in our manuscript – we went ahead and fixed them.

---

> > ### Comment · Reviewer_HPbv · 2021-11-30
> > **Response to authors**
> >
> > Thanks for your response. The main reason for my relatively low (positive) score despite acknowledging the paper's contribution is my low expertise on that topic (low confidence).

---

### Official Review · Reviewer_krSy · 2021-11-03

**Correctness:** 3
**Technical Novelty And Significance:** 3
**Empirical Novelty And Significance:** 3
**Recommendation:** 6
**Confidence:** 3

**Details Of Ethics Concerns:**

This paper focuses on the problem of user preference shift, which is caused by recommendation policies. Indeed, it may help to address some ethics concerns on personalized recommender systems.

**Main Review:**

Strengths:
1.	This paper focuses on an important problem that becomes increasingly important for recommender systems nowadays.
2.	The simulation framework, though with simplification and assumptions, is elaborately designed and seems promising for handling the preference shift problem.
3.	Idea of introducing users’ natural preference shift and penalizing the recommendation policy to stay in the trust-region is interesting and novel.

Weakness:
1.	Many technical details are missing, making it like a preliminary report.
-	In Sec. 4, when introducing estimating users’ preferences, the inference process is demonstrated in detail. On the contrary, descriptions about training a user predictive model given the historical data are vague, which is equally important.
-	For example, how to obtain $P(z^{\\pi '}_H | z^{\\pi }_\text{T+1})$ in Sec.4.1 (Estimation under known internal state dynamics)?
-	Eq. 3 seems to have a typo, "=" and "$\approx$" might swap their positions?
-	Authors have not released their experiment code, which makes it difficult for other colleagues to follow.
2.	Experiment details are also not clear. For example,
-	In ground truth human dynamics, given $b_t^H(s)$ and $u_t$, how is the distribution over users’ next timestamp preferences generated?
-	Does the counterfactual preference estimation used in evaluating the possible influence of recommendation policies?
-	What is the myopic method used in experiments?
-	What is the physical meaning of the “SUM” (of engagement)? Authors claim in the abstract that their framework can avoid manipulative behaviors but still generate engagement. What is the meaning of generating engagement? It seems to me that this framework can improve engagement under the initial or naturally shifted user preferences according to Table 1/2.
3. Presentation quality can be improved by polishing up the wording and rearranging the figures and tables in Sec. 6&7.


**Summary Of The Paper:**

This paper discusses an interesting but important problem in recommender systems, i.e., the users’ preference shift under the influence of recommender systems. Currently, both RL-based and myopic solutions cannot avoid influencing preferences in undesirable ways. Instead, authors provide a simulation framework that enables system designers to estimate the induced preference shifts an RS; evaluate its influence before deployment; and actively optimize to avoid such shifts. The basic idea is two-fold. The first design is to estimate user preference dynamics by training a user predictive model from historical user interaction data, which supports both future preference estimation and counterfactual estimation. The second one is to evaluate and optimize the current recommendation policy by measuring whether its influence is close to users’ natural preference shift.
By simulation study, authors show that recommender systems optimizing for staying in the above trust-region can avoid manipulative behaviors (e.g., changing preferences in ways that make users more predictable).


**Summary Of The Review:**

Although an interesting study, the paper has limitations (please see "weaknesses" section above). I would say that the current version of the paper is marginally below the acceptance threshold, but I am looking forward to the authors addressing my concerns above in their rebuttal.

## After Author Response
The authors' response has addressed most of my concerns and I choose to raise the score to 6.

Also, I expect the authors to format their released experiment code, such that followers can better understand the model and simulation.

---

> ### Author Response · Authors · 2021-11-14
> **Response to Reviewer krSy, Part 1**
>
> Thank you so much for your thoughtful review, and recognizing the importance of this problem.
>
> Despite the score of 5, we were encouraged that most of the points you raised were clarification questions. We attempt to answer them here, and will shortly update the manuscript to reflect these answers. Please let us know if the clarifications make sense.
>
> ----
>
> 1.a: We will make this clearer in the manuscript, but the "user predictive model" and the "preference inference process" are one and the same. As seen in Figure 2, the model used to estimate user's future preferences also outputs distributions over choices (predictions over the users' actions, the "choice belief"). In light of that, all the detail in Sec. 4 about the preference estimation immediately apply also to the behavior prediction model. Regarding details of the training process itself, the dataset of interactions is generated as described in section Appendix C.3, details about the model are in Appendix C.2, and the supervision signal is obtained as described in Figure 2 and Sec. 4.
>
> 1.b: To clarify, the expression $P(z_H^{\pi'} | z_{T+1}^\pi)$ is not used by our predictive model method (which is described in "Estimation under **unknown** internal state dynamics") but only by our oracle “baseline”/gold standard: the NHMM inference of future internal states when assuming that the user internal state dynamics are known. In brief, $P(z_H^{\pi'} | z_{T+1}^\pi)$ is just a standard inference in the NHMM. This is most easily understood in an equivalent HMM case, where this inference would consist of inferring the hidden state at some point in the future $h_H$ given a known starting state $h_0$, without any observations. The belief over the initial state would be a dirac distribution (with 1 at $h_0$). We would then propagate this belief to the next timestep using the known dynamics of the hidden state $P(h_1 | h_0) = \sum_{h_1} P(h_1 | h_0) P(h_0)$, and repeat until we reach the desired timestep ($P(h_t | h_0) = \sum_{h_t} P(h_t | h_{t-1}) \dots \sum_{h_1} P(h_1 | h_0) P(h_0)$). In the case of an NHMM, the only difference is that the dynamics are time-dependent. We will update the manuscript for further clarity on this.
>
> 1.c: The current ordering is actually correct: for ≈ in Eq. 3, the approximation comes from the fact that we are only considering the initial preferences rather than the full initial state. Considering the full internal state, i.e. $P(u_T^{\pi'} | x_{0:t}^\pi, s_{0:t}^\pi) = \int_{z_0} P(u_T^{\pi'} | z_0) P(z_0 | x_{0:t}^\pi, s_{0:t}^\pi)$, would constitute equality. Given that we can only recover an estimate over the initial preferences, we approximate the expression by $\int_{z_0} P(u_T^{\pi'} | u_0) P(u_0 | x_{0:t}^\pi, s_{0:t}^\pi)$. Note that this approximation is only required for counterfactual preference estimation, and constitutes the best estimate for the counterfactual preferences given the set of assumptions we have (doing better would require assuming other things about the user internal state). Regarding the equality in Eq. 3, this follows from the fact that the expression $\int_{u_0} P(u_T^{\pi'} | u_0) P(u_0 | x_{0:t}^\pi, s_{0:t}^\pi)$ is equivalent to calculating the probability of $u_T^{\pi'}$ by conditioning on the belief over the initial preferences $b^\pi_{0:t}(u_0) := P(u_0 | x_{0:t}^\pi, s_{0:t}^\pi)$. With an abuse of notation, we denote this as $P\big(u_T^{\pi'} | b^\pi_{0:t}(u_0) \big) := \int_{u_0} P(u_T^{\pi'} | u_0) P(u_0 | x_{0:t}^\pi, s_{0:t}^\pi)$. This will be made clearer in the revision.
>
> 1.d: We will have a public release of the experiment code once it's better cleaned and documented. [Here](https://drive.google.com/drive/folders/1Lqh1z6jAZ3xlAenIw3wSdXv5kCnzylZb?usp=sharing) is a link for the current preliminary version in case you would find it useful.

---

> ### Author Response · Authors · 2021-11-14
> **Response to Reviewer krSy, Part 2**
>
> 2.a: See the "Preference shift model." section of Appendix C.1. We will add more detail to it in the revised version. Intuitively, the user will be selecting preferences which they expect will lead them to be engaged in the future. In imagining the future, the user will go by their belief of future slates, and this is where $b_t^H(s)$ comes in.
>
> 2.b: Say you have a policy π you are interested in. Counterfactual preference estimation:
> - Is not required to estimate the future preferences that you would expect existing users to develop if you were to deploy π going forward
> - Is required to infer the preferences that you would expect new users, unbiased by your previously deployed policy, to develop.
>
> This is essentially the basis of our distinction between section 4.1 and 4.2. We will try to make this more clear in the manuscript.
>
> 2.c:
> > Authors claim in the abstract that their framework can avoid manipulative behaviors but still generate engagement. What is the meaning of generating engagement?
>
> When we talk about “engagement” broadly, we refer to $r_t(u_t^{\pi'})$ – the immediate reward that recysystems tend to optimize for today, i.e. showing content that users will want to consume; this is important in real world recommender systems, because very low engagement performance would indicate an unwillingness of users to use a system: e.g. even though a random recsys would probably do very well at not manipulating users, nobody would want to use such a system.
>
> > What is the physical meaning of the “SUM” (of engagement)?
>
> The simple engagement metric $r_t(u_t^{\pi'})$ (which is generally used in practice) does not penalize in any way preference changes – such metric is _preference-change-agnostic_: even if the system were to completely overwrite a user’s preferences, as long as the user is engaged, the system would be “performing well” under this metric. As some metrics which are not preference-change-agnostic, we introduce $r_t(u_0, \pi')$ and $r_t(u_t^{\pi_{\text{rnd}}}, \pi')$. However, these are not exactly what we care about either: we don’t want a system that actively tries to keep the user preferences static (as would result from blindly optimizing $r_t(u_0, \pi')$). Additionally, as we can’t perfectly define Natural Preference Shifts, it would also be incorrect to solely optimize to what a random recommender system would do.
>
> By taking the sum of these metrics, we are trying to lead the system to perform well under a variety of relatively-reasonable goals for the system, some of which are _not_ preference-agnostic, and thus actively penalizing preference shifts.
>
>
> > It seems to me that this framework can improve engagement under the initial or naturally shifted user preferences according to Table 1/2.
>
> If you are referring to the fact that the other metrics $r_t(u_0, \pi')$ and $r_t(u_t^{\pi_{\text{rnd}}}, \pi')$ increase when using a penalized RL system, that is right: this is part of the goal. The intention is that by ensuring that engagement would be high also for other preferences, the system will have less incentive to manipulate the user’s preferences arbitrarily.
>
> 3. Thanks for the suggestion! We’ll do so!

---

### Official Review · Reviewer_kWQ2 · 2021-11-07

**Correctness:** 2
**Technical Novelty And Significance:** 2
**Empirical Novelty And Significance:** 2
**Recommendation:** 5
**Confidence:** 4

**Main Review:**


## Strengths
1. The studied problem is interesting. Estimating the induced preferences are quite important for recommender systems, especially for real-world applications, where users indeed interact with recommendation algorithms.
2. The proposed method makes sense. It is well motivated.

## Weaknesses

- Why don't the authors deploy some wide-used simulators, such as Virtual-Taobao [1]. The literature survey about the simulator in recommender systems should be considered.
- It is always noisy if the experiments are based on the simulation.
- The experimental setup is overall too ideal. For example, in real-world applications, the recommendation engine/system may do not allow preference manipulation.
- There are a lot of choices that are selected biasedly. For example, the authors use a BERT model (Sun et al., 2019) as the human model, which leaves me a question: how about the other choices? Is this model the best choice?
- The authors should re-think the value of this work. If there is no solid evaluation manner, a proper solution is to plug this method into some commonly-accepted recommendation pipelines. For example, you can first use the estimated preferences shifts to improve the backbone recommendation models and then evaluate based on traditional evaluation manners of recommender systems. In the current version of this work, it is still unclear whether the method work or not.
- There is no competitive baseline method in the results (Table 1 and Table 2). I suggest the authors at least add some heuristic methods as baselines.
- The complexity of the proposed approach is unclear. Will it be a burden for the recommendation model?

[1] Shi, Jing-Cheng, et al. "Virtual-taobao: Virtualizing real-world online retail environment for reinforcement learning." Proceedings of the AAAI Conference on Artificial Intelligence. Vol. 33. No. 01. 2019.

## After Rebuttal
The authors have carefully summarized my concerns into five aspects. I agree with the reply to the 1st, 2nd, and 4th, while I partly agree or disagree with the 3rd and 5th. Thus I have raised my score.

**Summary Of The Paper:**

This paper studies an interesting problem of estimating preference shifts induced by recommender systems and proposes a distance metric for the shifted preferences based on the so-called "safe shifts". The proposed metric can help penalize undesired preference shifts, which can be easily used in training and evaluation. Experiments via the simulated recommendation environment and simulated human behavior verify the effectiveness of the proposed approach.


**Summary Of The Review:**

Overall speaking, this paper studies an interesting problem of estimating preference shifts induced by recommender systems. Nevertheless, the proposed method and experimental setup are based on too many assumptions, which may do not hold in the real world. Therefore, I recommend rejection.

---

> ### Author Response · Authors · 2021-11-14
> **Response to Reviewer kWQ2 Part 1**
>
> We deeply appreciate the time you took to review our work and your suggestions for improvement.
>
> Below we address the concerns raised in your review:
>
> ----
>
> **VALUE OF THE WORK WITHOUT TESTING IN THE REAL WORLD:**
>
> While we wholeheartedly agree that the end goal of this line of work would be to test these methods on real systems, there are many challenges that cannot be fully addressed in the scope of a single work.
>
> We’d like to emphasize that our main contribution is the idea that we should be attempting to anticipate preference shifts before deployment of a new policy, measure to what extent they are undesirable, and even actively optimize policies to avoid undesirable shifts (especially in light of the incentives RL recsystems will have to manipulate user preferences, and in light of the fact that making them myopic does not eliminate undesirable shifts). We provide a proof of concept that this could work, and we acknowledge that this is only a first step, tested in an idealized setting with relatively strong assumptions. However, we hope this can be a starting point for further research in this direction which focuses on relaxing such assumptions and making this method applicable to the complexity of real recommenders. To do so, we’d have to deal with obtaining realistic user choice models, choosing setting-specific convincing operationalizations of natural preference shifts, and addressing potential issues such as generalization and compounding errors, which is an entire research agenda and not something a single paper can hope to accomplish.
>
> However, this doesn’t mean that our results do not have value: we propose a framework on which the community can make progress, and show evidence that the direction is promising despite the problem’s tremendous difficulty.
>
> While we fully agree that our simulated evaluations do not speak directly as to whether the effects we talk about happen in real world systems, we still believe that our results still validate that our methodology can work and thus have value, as explained in the “Why simulation?” heading of Section 6.
>
> Additionally, although not fully practical, we believe that our work is an improvement relative to the vast majority of works that have tackled these kinds of problems previously – which make even stronger assumptions than ours (e.g. the full preference model to be known) and base their analyses entirely on such assumptions (see Section 2, “RS effects on users’ internal states”). We at least provide hope that, if the choice model is not hopelessly mis-specified, one can leverage data to make this method applicable and successful in the real world, with real users.
>
> ---
>
> **SIMULATORS AND VIRTUAL TAOBAO:**
>
> Regarding your point about simulators, in our related work section (under “Neural networks for recommendation and human modeling”) we acknowledge that simulating user interactions has been done before for the purposes of RL training. Additionally, we cite SIREN by Bountouridis et al., 2019, which attempts to estimate recsys effects on users in a simulated recsys environment, but with hardcoded user models (rather than learned). We were not aware of Virtual-Taobao, which is a much more elaborate and real-world-scale recsys environment simulation and also learns human models from data – and have updated our manuscript to cite it. If you have other recommendations as to related work regarding the simulator in recsys, we would gladly incorporate it.
>
> While Virtual-Taobao is very valuable in general, it is not the right platform for us to test our idea, for 3 reasons: 1) One thing we are particularly interested in are the preference changes of a user over potentially longer interaction sequences: our understanding is that Virtual Taobao would be unable to simulate how users’ requests would change as influenced by policies, as users can only be simulated for a single results browsing session. 2) More generally, we would not expect user preferences to change much based on the interaction with the recommender system in the e-commerce setting – especially within the context of a single request to the system (while browsing pages of results). 3) (less importantly, but a mismatch nonetheless in assumptions) Virtual Taobao's action space does not conform to the simple slate-choice formalism (at every timestep, the user does not have to choose an item from the slate shown to them by the recsys), meaning that our method would have to be extended before it could be applied to this setting.
>
> ----
>
> Response continued below

---

> > ### Comment · Reviewer_kWQ2 · 2021-11-29
> > **Reply to authors one by one**
> >
> > 1. Thanks. I agree that it is a challenging task that cannot be well handled by one paper. I suggest clarifying the contributions and limitations.
> > 2. Thanks. These three reasons for not using virtual-taobao make sense.
> > 3. Partly agree with that. Maybe more experiments with diverse model choices will make your statement more solid.
> > 4. Agree with that.
> > 5. Disagree with that. Training time is equally important to inference time for recommender systems.
> >
> > Overall, thanks for answering my questions. Therefore, I have raised my score.

---

> ### Author Response · Authors · 2021-11-14
> **Response to Reviewer kWQ2 Part 2**
>
> **MODELING CHOICES:**
>
> > There are a lot of choices that are selected biasedly. For example, the authors use a BERT model (Sun et al., 2019) as the human model, which leaves me a question: how about the other choices? Is this model the best choice?
>
> Firstly, as we discuss under the “Estimation under unknown internal state dynamics” heading in Section 4.1 and 4.2 and also show in more detail in Appendix A, our modeling assumption of using a sequence model is motivated by the fact that our method approximates tasks of interest in the NHMM equivalent to our problem setting. In that sense, we have carefully and intentionally chosen our model form in order to approximate such tasks.
>
> Regarding our specific choice of which sequence model to use, it is not our intention to claim that BERT is the best possible model for the task (as mentioned in the paper, any sequence model could work). We simply think it's a reasonable choice: one advantage of using the BERT architecture is that it allows for the possibility of training and using one single model to perform all inference tasks we are interested in in this domain: inferring initial preferences based on future context, future preferences based on earlier context, etc. Additionally, transformer models have been shown to do well in practice in sequential decision tasks relative to classic sequence models such as RNNs, further supporting this choice. In light of this, we would disagree that is a biased choice.
>
> ---
>
> **BASELINES:**
>
> Regarding baselines, the myopic and RL approaches with no penalization can be thought of as baselines, in that these approaches are similar to ones currently deployed in real-world practice. Our proposed method (penalized RL) performs differently. In terms of recovering counterfactual user preferences, there are no previous works we are aware of that would be applicable in our domain which we could baseline with.
>
> ---
>
> **OVERHEAD FOR RECOMMENDER SYSTEMS:**
>
> Regarding the complexity of our approach to train a penalized recommender policy: yes, it is harder to do this than to just optimize for engagement, but we’re trying to address the fundamental problem that recommender systems might be manipulating users, and we most certainly can’t expect the solution to be trivial. Further, while there is an overhead at training time (due to the simulations required), once the recsys model is trained there is no additional overhead in complexity relative to standard policies used in real-world systems. We’re happy to provide more details if desired.

---

### Official Review · Reviewer_vNt7 · 2021-12-02

**Correctness:** 3
**Technical Novelty And Significance:** 3
**Empirical Novelty And Significance:** Not applicable
**Recommendation:** 6
**Confidence:** 4

**Main Review:**

I do not wish to reiterate the strengths and weaknesses as described by the other reviewers.  Instead, I will pose a set of points for the authors to consider as they revise the paper.

- Is it true that consumer preferences shift?  In support of this claim, the authors cite models that assume that consumer preferences shift.  There could be many reasons for shifting preferences; for example novelty-seeking or learning one's own preferences are two distinct cases.  In other situations, preferences might not shift at all.  Citing work outside of computer science might be the best for supporting this point [e.g., 1].
- More subtly, there is a difference between consumer preferences and consumer behavior [2]. This paper does not distinguish between the two.
- Even when consumer preferences shift, do RSs really influence users?  Prior work with Amazon data suggests that at least 75% of observed activity would likely occur in the absence of recommendations [3].
- I am fully supportive of simulation methods; this is an active area of research, however, and best practices are still being established (see the "Workshop on Simulation Methods for Recommender Systems," https://simurec.piret.info/).  Figure 1 needs to point to the assumptions used to generate it (otherwise it might as well be a cartoon).  I strongly encourage the release of simulation source code to ensure that the work is reproducible and that other researchers can explore performance under modified simulation assumptions.  Additionally, showing similar results under different simulation assumptions would go a long way to dispel concerns about the use of simulation methods.
- There are many LaTeX formatting issues that need to be fixed.  E.g., tilde over 30% and 45% on page 2, backwards quotes all over the place, etc.

[1] Dzyabura, Daria and Hauser, John R., Recommending Products When Consumers Learn Their Preferences (February 15, 2017). Available at SSRN: https://ssrn.com/abstract=2202904 or http://dx.doi.org/10.2139/ssrn.2202904

[2] Friese, Malte, Michaela Wänke, and Henning Plessner. "Implicit consumer preferences and their influence on product choice." Psychology & Marketing 23.9 (2006): 727-740.

[3] Sharma, Amit, Jake M. Hofman, and Duncan J. Watts. "Estimating the causal impact of recommendation systems from observational data." Proceedings of the Sixteenth ACM Conference on Economics and Computation. 2015.





**Summary Of The Paper:**

In this paper, the authors argue that it is important to 1) estimate the impact of recommendation systems of user preferences, 2) evaluate if the shifts would be undesirable, and 3) optimize to avoid undesirable shifts.  The authors propose a method to do this and rely on simulations to evaluate it.

**Summary Of The Review:**

I think this paper could be modified to 1) better motivate the work and 2) consider alternative simulation assumptions to show the method is robust.  Otherwise, it addresses an interesting problem in a new way.

---

### Author Response · Authors · 2021-11-23
**Additional Feedback**

Dear reviewers, with the discussion period coming to an end, could you please confirm whether our responses above have addressed your main concerns or whether there are any remaining concerns and how we could address them? Thanks!

---

### Decision · Program_Chairs · 2022-01-20

**Decision:**

Reject

**Comment:**

This paper studies the influence of recommender systems on users' preferences. The authors propose a method for estimating preference shifts, evaluating their desirability, and avoiding such shifts (when needed).

After the initial review and discussion period, a fourth reviewer with significant recsys experience and a very good knowledge of this sub area was invited to provide an additional review of the paper. This is reviewer vNt7. Their review was positive overall but did highlight some limitations and potential ways to improve the paper's grounding in the recsys literature.

Overall, the main strengths of this paper were that it studies an interesting and practically motivated question. The reviewers also found the proposed solution reasonable.

The main limitations are twofold. One, the results use a single set of simulation assumptions. Showing similar results under different simulation assumptions would be helpful to better understand the robustness and potential limitations of the approach. Two, there is a certain disconnect with the simulation literature. See comments from reviewers vNt7 and kWQ2 (although I found your reply to Virtual-Taobao convincing).

Overall and given the final reviewer recommendations (three marginally above and one marginally below), this is a very borderline paper. However, the consensus view of the committee is that it would benefit from additional work before publication.

I am sorry that I cannot recommend acceptance at this stage. I do believe that some of the suggestions from the reviewers highlighted above (more diverse simulation, better grounding in current recsys simulation literature and in the field) will be useful in preparing the next version of this work.